# Levels of Physical Activity and Mental Health in Adolescents in Ireland

**DOI:** 10.3390/ijerph18041713

**Published:** 2021-02-10

**Authors:** Michal Molcho, Aoife Gavin, Devon Goodwin

**Affiliations:** 1School of Education, National University of Ireland Galway, H91 TK33 Galway, Ireland; devon.goodwin@nuigalway.ie; 2Health Promotion Research Centre, School of Health Sciences, National University of Ireland Galway, H91 TK33 Galway, Ireland; aoife.gavin@nuigalway.ie

**Keywords:** physical activity, exercise, mental health, adolescent health

## Abstract

The benefits of physical activity for the physical health of individuals are well documented. Less is known about the benefits of physical activity for mental health. This paper explores the associations between physical activity and positive mental health and mental health problems. The paper utilises data collected from a representative sample of 10–17-year-old adolescents in Ireland. Physical activity in the study is measured using moderate-to-vigorous physical activity (MVPA) and vigorous physical activity (VPA). Mental health was measured using the Cantril Leader of Life Satisfaction, the WHO-5 index, Mental Health Inventory (MHI-5) and the Health Behaviour in School-Aged Children (HBSC) Symptom Checklist (HBSC-SCL). Data were analysed using bivariate (Pearson Correlation, t-test, one-way ANOVA) and multivariate (two-way ANOVA, ordinary least squares (OLS) regressions) analyses. In total, 8636 adolescents were included in this analysis. Higher participation in physical activity was associated with higher scores on the positive mental health indicators and lower scores on the mental health problems indicators. When modelled together, VPA was a stronger predictor of mental health than MVPA, especially in girls. For example, standardised beta coefficients for predicting MHI-5 were −0.09 for MVPA (*p* < 0.001) and −0.13 for VPA (*p* < 0.001) To our knowledge, this is the first study that looks at levels of physical activity as well as both positive mental health and mental health problems. The study highlights the need to encourage and enable adolescents, and especially girls, to participate in vigorous exercising as way of promoting positive mental health.

## 1. Introduction

The benefits of physical activity to physical health are well established [1,2,3]. Physical activity is an essential component of a healthy life, and a preventative measure against coronary heart disease, diabetes, hypertension, as well as a measure promoting cardio-respiratory health and bone and joint health [4,5,6]. In fact, the documented physical benefits of physical activity are such that physical activity, or lack thereof, is considered a major public health concern [7]. While the evidence of physical activity on physical health and wellbeing is well demonstrated, less is known about the correlations between physical activity and mental health. To help addressing this gap, the current paper uses two measures of physical activity that are based on the WHO recommendations on the level of activity that children and adolescents should achieve [5] and examines their correlations with four established measures of wellbeing and mental health in an adolescent population.

Measuring the impact of physical activity on mental health is of great importance, as mental health concerns are rising and physical activity levels remain lower than the recommended levels in adolescent populations [7]. Evidence suggests that up to one in five adolescents report mental health problems, and that the frequency of mental health concerns in developed countries is increasing [8,9]. At the same time, the vast majority of people of all ages, and especially women and girls, do not meet the recommended levels of physical activity [7,10]. If our study does, indeed, support the hypothesis that higher levels of physical activity increase positive mental health and decrease mental health problems, then it can offer a relatively cheap, yet powerful, intervention, with benefits that go beyond just improving mental health. 

In order to fully assess the impact of physical activity on both positive and negative mental health, it is important to use well-validated measures. While positive mental health, in the form of life satisfaction, wellbeing and quality of life are often tested and reported in adolescents [11,12,13,14], the use of well-validated indicators of mental health problems, or negative mental health, is relatively limited, especially in nationally representative population surveys. To address this gap in the literature, we used two indicators of positive mental health and wellbeing and two indicators of negative mental health and mental health problems. For positive mental health and wellbeing, we used the Cantril Life Satisfaction Ladder, a well-tested and validated indicator of global wellbeing [15,16], and the World Health Organization Wellbeing Index (WHO-5) [17,18,19]. We measured negative mental health and mental health problems using the Health Behaviour in School-Aged Children (HBSC) Symptom Checklist (HBSC-SCL) [14,20] and the Mental Health Inventory (MHI-5) [21,22] which is a brief, useful [23], sensitive [24] and valid [25] mental health inventory for assessing mental health problems in adolescents [22,26].

The two indicators of physical health considered in the current study are those developed based on the WHO recommendations and which measure moderate-to-vigorous physical activity (MVPA) and vigorous physical activity (VPA). The WHO recommends that children and adolescents aged 5–17 should participate in MVPA at least 60 min a day and engage in muscle-building activities (sometimes measured through VPA) 2–3 times per week [5]. Much of the research on physical activity in adolescents has utilised one of these two indicators, with MVPA seeming to predominate the literature. Importantly, however, these two indicators are often examined in isolation from each other, even though the literature does recognise the importance of measuring both MVPA and VPA given their separate contribution to physical health and wellbeing [8,10,11], and given that VPA and MVPA do not necessarily overlap [8,11,27]. To address this, the current study examines both behaviours together to fully appreciate the contribution of different types and intensity of physical activity and to establish their benefit to adolescent mental health, both separately and over and above one another. 

Specifically, using a population survey, this paper explores the impact of physical activity—both MVPA and VPA—on positive mental health and on mental health problems, aiming to identify the impact that the different intensity of physical activity has on both positive and negative mental health, thus addressing this gap in the existing literature. Modelling the levels of intensity of physical activity allows us to determine what level of intensity is most impactful for mental health. Segregating the analyses by gender allows us to identity the differential impact of physical activity among boys and girls. 

## 2. Method

### 2.1. Procedure and Sample

The Health Behaviour in School-Aged Children (HBSC) is a cross-sectional study conducted every four years as part of the World Health Organization (WHO) collaborative HBSC study [28], with nationally representative samples of children and adolescents in Europe and Canada. In Ireland, a two-stage sampling process was used with schools as the primary sampling unit. In the first stage, primary and post primary schools were randomly selected from the national total across eight geographical regions. At the second stage, individual classrooms within participating schools were randomly selected to partake. Schools were recruited during school term time between April 2018 and October 2018. Participating children and adolescents were asked to complete a self-reported questionnaire within the classroom including questions on health outcomes, health behaviours and socio-demographic factors. Only children and adolescents who provided informed consent took part in the study. Ethical approval was granted from the Research Ethics Committee, NUI Galway, Ireland. 

The response rate at the school level was 63%, and the response rate at the student level was 84%, resulting in a sample of 12,002 participants, 10 to 17 years of age, of whom 51% of participants were girls, and the average age was 13 years of age. For the purpose of this paper, we used a subset of the data that included participants that had complete data on gender, age, physical activity and mental health questions. The sample for this paper consists of 8636 participants, of whom 53% are girls, and the mean age of participants is 14 years of age.

### 2.2. Measures

#### 2.2.1. Moderate-to-Vigorous Activity (MVPA)

Moderate-to-vigorous activity (MVPA) was measured with following item: “Physical activity is any activity that increases your heart rate and makes you get out of breath some of the time. Physical activity can be done in sports, school activities, playing with friends, or walking to school. Some examples of physical activity are running, brisk walking, rollerblading, biking, dancing, skateboarding, swimming, soccer, basketball, football, and surfing. Over the past 7 days, on how many days were you physically active for a total of at least 60 min per day? Please add all the time you spent in physical activity each day.” Response options ranged from 1—none of the days to 8—seven days.

#### 2.2.2. Vigorous Physical Activity (VPA)

Vigorous physical activity (VPA) was measured with the following two items. The first item asked about the frequency of vigorous activity: “Outside school hours: How often do you exercise in your free time so much that you get out of breath or sweat?” Response options were 1—never, 2—less than once a month, 3—once a month, 4—once a week, 5—2 to 3 times a week, 6—4 to 6 times a week and 7—every day. The second item asked about hours of vigorous activity in a usual week: “Outside school hours: how many hours a week do you usually exercise in your free time so much that you get out of breath or sweat?” Response options were 1—none, 2—about half an hour, 3—about 1 h, 4—about 2 to 3 h, 5—about 4 to 6 h, 6—about 7 h or more. To create a single indicator of VPA based on the two single items, we multiplied the two items [29,30,31]

#### 2.2.3. Life Satisfaction

Life satisfaction was measured using the Cantril Ladder for general life satisfaction [15]. Minor changes in wording were conducted on the original item to facilitate its use with 11-year-olds, and this revised version was piloted in five countries in spring 2001 [16,28]. The question is presented pictorially as a ladder of 11 steps from 0 to 10 and introduced by a text describing 10 to be indicating the “best possible life” and 0 to indicate “the worst possible life” for you. Participants were asked: “Here is a picture of a ladder. The top of the ladder ‘10′ is the best possible life for you and the bottom ‘0′ is the worst possible life for you. In general, where on the ladder to you feel you stand at the moment?”

#### 2.2.4. WHO-5 Wellbeing Index

The WHO-5 was originally presented at a WHO meeting in 1998 as part of a project on the measurement of wellbeing in primary health care patients [18] and since then has been frequently used in clinical settings [18], including on children and adolescent populations [32]. WHO-5 consists of 5 items, where participants are asked to report how often in the last two weeks they felt the following: “I have felt cheerful and in good spirits”; “I have felt calm and relaxed”; I have felt active and vigorous”; “I woke up feeling fresh and rested”; and “My daily life has been filled with things that interest me”. The response options for all these questions were: 1—at no time, 2—some of the time, 3—less than half of the time, 4—more than half of the time, 5—most of the time, 6—all of the time. The raw score for the scale was calculated by totalling the figures of the five answers. The raw scores ranged from 0 to 25, 0 representing worst possible and 25 representing best possible quality of life. To obtain a percentage score ranging from 0 to 100, the raw score was multiplied by 4. A percentage score of 0 represents worst possible, whereas a score of 100 represents best possible quality of life [17]. In this paper, we used the percentage score of the scale.

#### 2.2.5. HBSC Symptom Checklist (HBSC-SCL)

The HBSC Symptom Checklist (HBSC-SCL) was introduced to the HBSC survey in the 1993/1994 round of data collection [33,34,35] and has been widely tested and validated for assessing emotional and physical wellbeing [14,20,35,36]. Participants are asked to report how often in the past 6 weeks they had the following: headache, stomachache, backache, feeling low, irritability or bad temper, feeling nervous, difficulties in getting to sleep and feeling dizzy. The response options for all of these eight items are 1—rarely or never, 2—about once a month, 3—about once a week, 4—more than once a week and 5—about every day. These items were used as a summative scale [20]. The validity of the scale in our study (Cronbach α = 0.845) confirmed the psychometric properties of this scale as was previously reported [20,35,36].

#### 2.2.6. Mental Health Inventory (MHI-5)

The Mental Health Inventory (MHI-5) is a brief 5-item instrument that measures general mental health [22,26] and is a subscale of the Short Form Health Survey (SF-36) [37]. The instrument contains the following questions: “How much of the time during the last month have you: been a very nervous person; felt downhearted and blue; felt calm and peaceful; felt so down in the dumps that nothing could cheer you up; and been a happy person?”. The response options of these five questions are: 1—all of the time (1 point), 2—most of the time (2 points), 3—a good bit of the time (3 points), 4—some of the time (4 points), 5—a little of the time (5 points) and 6—none of the time (6 points). The scoring of the two items that ask about positive feelings was reversed. The score for the MHI-5 was computed by summing the scores of each question item and then transforming the raw scores to a 0–100 point scale [37]. 

#### 2.2.7. Socio-Demographic Characteristics 

Participants were asked to report their gender as boy or girl. The age of each adolescent was calculated using the survey administration date and self-reported month and year of birth. Age group categories were created and were 10–11 years, 12–14 years and 15–17 years. Age categories are used to report the distribution of physical activity and mental health indicators, while real age was used in ordinary least squares (OLS) regression models. Participants were also asked to record whether their mother and/or father have a job, where their parent(s) work and what exact job their parent(s) have. From these data, each parent was assigned to a social class group as professional managers, managerial, non-manual, skilled manual, semi-skilled and unskilled, and unknown/unclassified [38]. The higher reported level of parental social class was used if parental social classes differed. Social class was then further recoded into three groups as high (professional managers, managerial), medium (non-manual, skilled manual) or low (semi-skilled and unskilled). 

### 2.3. Statistical Analysis

Analyses were carried out in SPSS 26.0 (IBM Corp., Armonk, NY, USA). Means of mental health and physical activity measures are presented by gender and age groups. Means were compared using *t*-test for gender, and one-way ANOVA for age groups to assess the statistical significance of the differences. A two-way ANOVA was used to assess the interaction between gender and age on all mental health and physical activity measure. The correlations between the different levels of physical activity and mental health were tested through Pearson correlations coefficients and are presented by gender and age groups. Lastly, to test the impact of the different levels and intensity of physical activity on mental health, we used OLS regression models, segregated by gender and controlled for social class. Effect sizes of the correlations coefficient and standardised beta coefficient were interpreted using Cohen’s operational guidelines for interpreting coefficients in large size studies, with r = 0.10 considered small effect; r = 0.30 considered medium effect; and r = 0.50 considered large effect size [39].

## 3. Results

Overall, 8636 adolescents in Ireland aged 10–17 were included in the analyses for this paper. Table 1 and Table 2 present the mental health and physical activity profiles of this sample. Boys and younger adolescents reported more positive mental health outcomes, both in terms of life satisfaction and in terms of wellbeing, while girls and older adolescents reported poorer mental health (higher scores on the MHI-5) and more frequent negative physical and mental symptoms (higher scores on the HBSC-SLC). Equally, boys and younger adolescents reported higher participation in both moderate-to-vigorous physical activity and vigorous physical activity.

While the same patterns remain when analysing both mental health outcomes and physical activity indicators by age within gender, it also becomes clear that the differences between the age groups are greater among girls than among boys. For example, the mean score for boys on WHO-5 decreases from 69.91 (SD = 21.15) in the youngest age group to 56.20 (SD = 21.43) in the oldest age groups (*p* < 0.001), while in girls, the change is from 69.42 (SD = 21.93) among 10–11-year-olds to 54.64 (SD = 21.27) among 15–17-year-olds (*p* < 0.001). Essentially, we found that in most of the mental health outcomes, the baseline mean among 10–11-year-olds was quite similar among boys and girls, but girls fared worse than boys on all outcomes in the older age group (*p* < 0.01 for all two-way ANOVA analyses). The same pattern is evident with regard to physical activity; the level of activity among 10–11-year-olds is rather similar in girls and boys, but the level of activity on both indicators is much lower among 15–17-year-old girls compared to boys of the same age group (*p* < 0.001 for all two-way ANOVA analyses).

Table 3 presents the Pearson correlation coefficients between physical activity and mental health, by gender and age group. Overall, physical activity, both MVPA and VPA, was significantly correlated with positive mental health (*p* < 0.001). An increased frequency of participation in physical activity is correlated with increased scores on the two indicators of positive mental health (higher reported life satisfaction and wellbeing) and with decreased scores on the two indicators of negative mental health (MHI-5 and HBSC-SCL). The strongest correlation was between MVPA and WHO-5 in girls (r = 0.342; *p* < 0.001), while the weakest correlation was between VPA and MHI-5 among 10–11-year-olds (r = -0.07; *p* < 0.05). With the exception of life satisfaction, physical activity indicators were stronger predictors of positive mental health outcomes in girls compared to boys. Among 15–17-year-olds, the correlation between physical activity and all mental health outcomes was stronger for VPA compared to MVPA. In terms of the magnitude of the size effects reported in Table 3, they all vary from low to medium effect size [39,40].

Following from the bivariate analysis of physical activity and mental health, we proceeded to conduct multivariate analyses modelling the two physical activity indicators together to assess how they predict mental health outcomes (Table 4). Given the gender differences in both participation in physical activity and mental health, we ran the models for the whole sample first, and subsequently separately for girls and boys. In 9 out of the 12 models we ran, the effect of VPA on mental health was found to be greater than that of MVPA, although age was the strongest predictor of mental health over and above physical activity. The explained variability (adjusted R^2^) of the models varied from a low 0.040 (*p* < 0.001) in predicting symptoms among boys to low to medium 0.20 (*p* < 0.001) in predicting wellbeing among girls. Overall, the explained variability in all of the models was higher among girls than it was among boys. While the explained variability is not high, within the context of real life studies, and especially when studying such a complex phenomenon such mental health, the fact the physical activity can explain 10–20% of the change in mental health is substantial, especially in the context of interventions [40]. 

## 4. Discussion

This study examined the associations between physical activity and mental health, focusing on different levels of intensity of physical activity as predictors of both positive and negative mental health. We started by examining the mental health profile of adolescents in Ireland. Our data clearly demonstrated that boys and younger adolescents score higher on the indicators of positive mental health (the life satisfaction and the WHO-5 scale) and lower on the indicators on negative mental health (the symptoms checklist and the MHI-5). That boys and younger adolescents report better mental health has been widely reported previously internationally [41,42,43] and in Ireland [38,44,45], and the current findings support this. We also found that the decrease in reported positive mental health and increase in reported mental health problems in girls as they get older is more pronounced than that of boys, as was previously reported in Ireland [38,44]. We then examined the profile of participation in physical activity in Ireland and found that boys and younger adolescents scored higher on both the MVPA and the VPA scales, indicating a higher level of physical activity than girls and older adolescents, and once again, the decrease is more evident among girls as they get older than it is in boys. That, too, is in line with the literature [29,38,41]. 

Next, we examined the associations between each of the two physical activity indicators and each of the mental health indicators, for the whole sample, for boys and girls separately and for the three age groups. Our findings clearly indicate that across all groups, higher levels of participation in physical activity were associated with higher scores on the positive mental health indicators and lower scores on the negative mental health indicators, with correlation coefficients ranging from a low effect size r = −0.07 (*p* < 0.05) between VPA and MHI-5 among 10–11-year-olds, to a medium effect size of r = 0.34 (*p* < 0.001) between MVPA and WHO-5 in girls. Overall, physical activity was more strongly associated with positive mental health in girls compared to boys. This is consistent with the literature that suggests that sedentary behaviour is linked with poor mental health [46], even though sedentary behaviour should not be viewed as one end of the continuum with high physical activity being the other end [47,48]. The effect size of all the correlation coefficients included in these analyses varied from low to medium in terms of the magnitude of the effect, as described in Cohen’s operational guidelines for interpreting coefficients in large sizes studies [39], and is consistent with Hemphill’s suggestion that when measuring complex outcomes, such as mental health, such effect sizes are still of significance [40].

Lastly, we used models of ordinary least squares regressions to identify the impact of the different levels of physical activity on mental health, as well as the overall model fit for the whole sample and for boys and girls separately. In most cases, our models suggested that the impact of VPA on mental health was stronger than that of MVPA, suggesting that more strenuous exercise has a stronger and more positive impact on the mental health of adolescents. It was also clear from the model fit that the impact of physical activity was stronger in girls than in boys. The model fit ranged from a weak adjusted R^2^ = 0.04 (*p* < 0.001) in predicting symptoms among boys to a medium adjusted R^2^ = 0.20 (*p* < 0.001) in predicting wellbeing among girls. While the explained variability of the models is not high, it is not unusually low for outcomes such as mental health that are influenced by multiple factors in one’s life [39,40]. 

The benefits of physical activity to physical health and life satisfaction are well established [1,2,3,4], as well as the fact that the majority of children and adolescents do not achieve the recommended level of activity [7,29,41]. We also know that mental health problems are increasing [8,9,20], and that girls and older adolescents consistently report poorer mental health, lower life satisfaction and more negative mental health symptoms [20,41,42]. However, while we know that MVPA is associated with better life satisfaction and fewer negative mental health symptoms [1,2,49], less is known about the impact of VPA in mental health [2,50] or about the differential impact of VPA, over and above that of MVPA. Our findings clearly indicate that VPA is a stronger predictor of better mental health than MVPA and that the impact of physical activity on mental health is stronger in girls than in boys; given that girls report poorer mental health than boys, and that VPA has historically been given less attention in the literature, these findings are of importance. 

Our findings not only indicate the importance of physical activity for mental health, but we also identified that, especially among girls, interventions should start early. Studies suggest that as girls grow up, they tend to experience more barriers to physical exercise compared to boys, while boys present more motivation for the same [51]. The reasons that girls give for lack of interest in physical exercise include perceived lack of competence and insufficient time [51]. Perhaps more importantly, however, girls also report being concerned with teasing and harassment around appearance, as exercising is described as crossing traditional gender boundaries [52]. While we can only speculate as to whether this is the case for the participants in this study, we certainly do see the same pattern of decreased participation in physical exercise in older girls, coupled with poorer reported mental health. Suggested ways to address under-participation in exercise in adolescents include identifying the health benefits of exercising to increase internal motivation, while also creating an environment that sees health and exercise as valued by teens, parents, teachers and the media [53]. Adopting such an approach could help to improve the physical as well as mental health of both boys and girls.

### Strengths and Limitations

This study has a number of strengths. The Irish HBSC study is a large, nationally representative study. The items included in the questionnaire have been previously validated. Adolescent participants were asked to report on their mental health in various ways and on two different types of participation in physical activity. Although self-reported activity data may not provide accurate estimates, it is arguably appropriate for ranking individuals. Limitations to this study include that the indicators are subjective and self-reported. As this study is cross-sectional, temporal associations are possible, and causality cannot be inferred.

## 5. Conclusions

Older adolescents (compared to younger adolescents), and especially girls (compared to boys), report lower levels of life satisfaction and wellbeing and higher levels of negative mental health symptoms and problems. As the world is currently grappling with the COVID-19 pandemic, we can expect that more adolescents will struggle with their mental health [54]. Our findings suggest that one way of addressing this is through increasing participation in not only in physical activity in general, as was demonstrated in research conducted with healthcare workers [55], but in vigorous physical activity in particular. Creating opportunities to engage in physical activity and encouraging participation in sport is a relatively cheap and easy intervention, but for that to happen, there needs to be a concentrated effort at the national level, an effort that addresses adolescents, parents, teachers and the media. One implication of these findings is that it supports the proposal that physical activity should be part of the school curriculum in a more frequent and meaningful way [56,57], especially as children grow older. In addition, community spaces for participation in sport and physical activity should be created, and participation in sport, rather than just competition, needs to be recognised for its importance for the overall health of children and adolescents. More specifically, physical activity and sport should be seen as a key method of promoting positive mental health and preventing mental health problems in adolescence.

## Figures and Tables

**Table 1 ijerph-18-01713-t001:** Indicators of wellbeing and mental health by gender and age group.

Mental Health Indicators				
	Life Satisfaction	WHO-5 ^1^ Wellbeing	MHI-5 ^2^ Mental Health	HBSC-SCL ^3^ Symptoms
	M (SD)	M (SD)	M (SD)	M (SD)
All	7.44 (1.82)	57.87 (23.32)	29.62 (19.59)	16.51 (6.67)
Boys	7.54 (1.72)	61.39 (22.04)	25.59 (17.37)	15.06 (5.71)
Girls	7.35 (1.90)	54.82 (24.00)	33.13 (20.71)	17.77 (7.16)
	*p* < 0.001	*p* < 0.001	*p* < 0.001	*p* < 0.001
10–11	8.28 (1.64)	69.52 (21.53)	21.62 (15.81)	13.65 (5.29)
12–14	7.56 (1.80)	59.14 (22.93)	28.82 (19.30)	16.20 (6.57)
15–17	6.85 (1.74)	50.32 (22.00)	34.73 (20.18)	18.37 (6.84)
	*p* < 0.001	*p* < 0.001	*p* < 0.001	*p* < 0.001
Boys				
10–11	8.23 (1.60)	69.61 (21.15)	20.88 (14.73)	33.73 (5.57)
12–14	7.64 (1.72)	61.72 (21.78)	25.05 (17.52)	32.30 (6.16)
15–17	6.99 (1.62)	56.20 (21.43)	29.07 (17.83)	31.24 (6.06)
	*p* < 0.001	*p* < 0.001	*p* < 0.001	*p* < 0.001
Girls				
10–11	8.32 (1.68)	69.42 (21.93)	22.38 (16.84)	33.20 (6.28)
12–14	7.49 (1.87)	56.90 (23.67)	32.09 (20.16)	29.93 (7.31)
15–17	6.73 (1.83)	45.64 (21.27)	39.23 (20.81)	27.50 (7.17)
	*p* < 0.001	*p* < 0.001	*p* < 0.001	*p* < 0.001

^1^ World Health Organization-5. ^2^ Metal Health Inventory-5. ^3^ Health Behaviour in School-Aged Children Symptoms Check List. Gender differences were tested using *t*-test, and differences between age groups were tested using one-way ANOVA.

**Table 2 ijerph-18-01713-t002:** Indicators of physical activity by intensity, gender and age group.

Physical Activity Indicators		
	MVPA ^1^	VPA ^2^
	M (SD)	M (SD)
All	5.47 (1.97)	21.10 (10.80)
Boys	5.81 (1.90)	22.75 (11.00)
Girls	5.17 (1.98)	19.63 (10.45)
	*p* < 0.001	*p* < 0.001
10–11	6.36 (1.73)	22.95 (10.67)
12–14	5.63 (1.90)	21.40 (16.68)
15–17	4.79 (1.95)	19.73 (10.87)
	*p* < 0.001	*p* < 0.001
Boys		
10–11	6.53(1.65)	23.56(10.86)
12–14	5.90(1.85)	22.88(10.89)
15–17	5.26(1.95)	22.10(11.12)
	*p* < 0.001	*p* < 0.05
Girls		
10–11	6.19(1.80)	22.32(10.45)
12–14	5.39(1.92)	20.11(10.33)
15–17	4.43(1.88)	17.84(10.28)
	*p* < 0.001	*p* < 0.001

^1^ Moderate-to-vigorous physical activity. ^2^ Vigorous physical activity. Gender differences were tested using *t*-test, and differences between age groups were tested using one-way ANOVA.

**Table 3 ijerph-18-01713-t003:** Pearson correlation of physical activity and mental health by gender and age group.

Mental Health Indicators					
		Life Satisfaction	WHO-5 Wellbeing	MHI-5 Total	HBSC-SCL
All					
	MVPA	0.23 ***	0.33 ***	−0.23 ***	−0.21 ***
	VPA	0.19 ***	0.29 ***	−0.21 ***	−0.170 ***
Boys					
	MVPA	0.23 ***	0.28 ***	−0.19 ***	−0.12 ***
	VPA	0.19 ***	0.26 ***	−0.18 ***	−0.10 ***
Girls					
	MVPA	0.22 ***	0.34 ***	−0.22 ***	−0.22 ***
	VPA	0.20 ***	0.28 ***	−0.20 ***	−0.18 ***
10–11					
	MVPA	0.140 ***	0.18 ***	−0.12 ***	−0.11 **
	VPA	0.11 ***	0.20 ***	−0.13 ***	−0.07 *
12–14					
	MVPA	0.20 ***	0.30 ***	−0.18 ***	−0.16 ***
	VPA	0.21 ***	0.27 ***	−0.21 ***	−0.16 ***
15–17					
	MVPA	0.14 ***	0.28 ***	−0.19 ***	−0.15 ***
	VPA	0.17 ***	0.30 ***	−0.21 ***	−0.17 ***

* *p* < 0.05; ** *p* < 0.01 *** *p* < 0.001.

**Table 4 ijerph-18-01713-t004:** Models of ordinary least squares (OLS) regression predicting mental health by age and physical activity by gender.

	Life Satisfaction			WHO-5 Wellbeing			MHI5 Total			HBSC-SCL		
	All	Boys	Girls	All	Boys	Girls	All	Boys	Girls	All	Boys	Girls
Age	−0.26 ***	−0.24 ***	−0.27 ***	−0.24 ***	−0.19 ***	−0.29 ***	0.20 ***	0.15 ***	0.25 ***	0.22 ***	0.17 ***	0.27 ***
MVPA	0.09 ***	0.11 ***	0.06 **	0.17 ***	0.14 ***	0.16 ***	−0.09 ***	−0.08 ***	−0.07 ***	−0.08 ***	−0.04	−0.08 ***
VPA	0.11 ***	0.11 ***	0.12 ***	0.16 ***	0.17 ***	0.14 ***	−0.13 ***	−0.12 ***	−0.12 ***	−0.10 ***	−0.07 ***	−0.09 ***
N	8636	4017	4619	8636	4017	4619	8636	4017	4619	8636	4017	4619
Adjusted R^2^	0.12 ***	0.11 ***	0.13 ***	0.17 ***	0.13 ***	0.20 ***	0.10 ***	0.06 ***	0.11 ***	0.08 ***	0.04 ***	0.12 ***

** *p* < 0.01; *** *p* < 0.001. All models controlled for social class.

## Data Availability

The data presented in this study are available at http://www.nuigalway.ie/hbsc/dataaccess/ (accessed on 20 January 2021).

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
