# Peer review of "Levels of Physical Activity and Mental Health in Adolescents in Ireland"

_ijerph, 2021, doi:10.3390/ijerph18041713_

Round 1

Reviewer 1 Report

This manuscript entitled "Levels of physical activity and mental health in adolescents in Ireland" aimed to understand associations between physical activity and positive mental health and mental health problems.

The manuscript is very interesting. However, some issues should be addressed by the authors:

ABSTRACT

  • Incluse a short sentence about the statistical analysis
  • Include statistical results, as p value and predictor value when describing the results

INTRODUCTION

  • Please update the reference 5 (WHO, W., Global recommendations on physical activity for health. 2010).

METHODS

  • Lines 190 to 196 - Include how was interpreted the correlation results. For example: Strong by r values ranging from 0.7 to 1.0, moderate by r values 0.4 to 0.6, and weak by r values 0.1 to 0.3

RESULTS

  • Table 1 and Table 2: include the p value between age groups and sex. Without this test is not possible to make the affirmation in the lines 200 to 205
  • Before go direct to the relation between the outcomes Mental Health and Physical Activity, please, analyse more in-depth  each variable. For example, it is important for the reader to understand how was the pattern for Mental Health and for the Physical Activity, mainly by prevalence
  • Although the p value is quite significative in Table 3, most importante is the value from Correlation. Base on the Table 3, seems that almost all correlatin value are 'weak' (0.1 to 0.4). This should by addressed and carefully discussed by the authors. How this weak  relation support dos your conclusion? Are you interpreting correctly the results?
  • The same for the Table 4.

DISCUSSION

- For me, it is the main weakness. Please, improve the discussion section, include more recent articles specially from other countries (see below some sugestions) and carefully interpret your correlations results.

REFERENCES

  • Several recent articles from IJERPH could be cited. It will you be help you to compare the results with other countries.
  • Below I suggest some articles pre and during COVID pandemic about mental health in adolescents which may improve the introduction and discussion sections:

Escobar, D.F.S.S.; Noll, P.R.S.; Jesus, T.F.; Noll, M. Assessing the Mental Health of Brazilian Students Involved in Risky Behaviors. Int. J. Environ. Res. Public Health 202017, 3647.

Seven, Ü.S.; Stoll, M.; Dubbert, D.; Kohls, C.; Werner, P.; Kalbe, E. Perception, Attitudes, and Experiences Regarding Mental Health Problems and Web Based Mental Health Information Amongst Young People with and without Migration Background in Germany. A Qualitative Study. Int. J. Environ. Res. Public Health 202118, 81.

Tahara, M.; Mashizume, Y.; Takahashi, K. Coping Mechanisms: Exploring Strategies Utilized by Japanese Healthcare Workers to Reduce Stress and Improve Mental Health during the COVID-19 Pandemic. Int. J. Environ. Res. Public Health 202118, 131.

Escobar, D.F.S.S.; Jesus, T.F.; Noll, P.R.S.; Noll, M. Family and School Context: Effects on the Mental Health of Students. Int. J. Environ. Res. Public Health 202017, 6042.

Author Response

Thank you for your helpful review. Below you will find our response for each of your commenys. Many thanks for taking the time to revie wthe paper.

Comment 1:

ABSTRACT: Include a short sentence about the statistical analysis

Response 1:

The abstract now includes a sentence on the analyses performed in this study.

Comment 2:

Include statistical results, as p value and predictor value when describing the results

Response 3:

We included an example of our findings, including effect size and p value, however, due the work count limit of the abstract it is impossible to elaborate more on the findings in the abstract.

Comment 3:

INTRODUCTION: Please update the reference 5 (WHO, W., Global recommendations on physical activity for health. 2010).

Response 3:

We have now added to the introducation the revised WHO guidelines for physical activity:

Bull, F. C. et al., World Health Organization 2020 guidelines on physical activity and sedentary behaviour. British Journal of Sports Medicine 2020, 54 (24), 1451-1462.

Comment 4:

METHODS  Lines 190 to 196 - Include how was interpreted the correlation results. For example: Strong by r values ranging from 0.7 to 1.0, moderate by r values 0.4 to 0.6, and weak by r values 0.1 to 0.3

Response 4:

We have used Cohen’s operational guidelines for interpreting coefficient in large size studies, or as he states, in a more real life context. According to Cohen (1988), in terms of the magnitude of effect size, correlation coefficients should be interpreted in such way that r = 0.10 is considered small; r=0.30 is considered medium; and r=0.50 is considered large. We have now included text to this effect in the methods section.

Comment 5:

RESULTS Table 1 and Table 2: include the p value between age groups and sex. Without this test is not possible to make the affirmation in the lines 200 to 205

Response 5:

We have now added p values for the means in tables 1 and 2, as well as a breakdown by age and gender, and referred to these analyses in the results and in the discussion.

Comment 6:

Before go direct to the relation between the outcomes Mental Health and Physical Activity, please, analyse more in-depth  each variable. For example, it is important for the reader to understand how was the pattern for Mental Health and for the Physical Activity, mainly by prevalence

Response 6:

We do not feel that adding the prevalence will add to the paper, and perhaps will even add more confusion. To present prevalence, we will need to introduce new, dichotomous, variables that are not in line with the current analysis. We trust that the inclusion of ANOVA tests and gender X age analyses in tables 1 and 2, are providing more in-depth analyses for each variable, responding to this comment while still maintaining the coherence of the overall analyses of the paper.

Comment 7:

Although the p value is quite significative in Table 3, most importante is the value from Correlation. Base on the Table 3, seems that almost all correlatin value are 'weak' (0.1 to 0.4). This should by addressed and carefully discussed by the authors. How this weak  relation support dos your conclusion? Are you interpreting correctly the results?

Response 7:

The reviewer is, of course, correct that in the context of some psychological studies, such correlations would be considered weak. However, we are using Cohen’s operational guidelines for interpreting coefficient in large size studies, or as he states, in a more real life context. According to Cohen (1988), in terms of the magnitude of effect size, correlation coefficients should be interpreted in such way that r = 0.10 is considered small; r=0.30 is considered medium; and r=0.50 is considered large. Looking into this guideline critically, Hemphill (2003) suggested that even these guidelines maybe too high in some contexts and proposed to judge the effect size in the context of the complexity of the outcomes measured. We have included in the discussion text to that effect, as well as used Cohen’s guidelines for interpreting our and presented guidelines for interpretation in the results

Comment 8:

The same for the Table 4.

Response 8:

We did the same for table 4

Comment 9:

DISCUSSION

- For me, it is the main weakness. Please, improve the discussion section, include more recent articles specially from other countries (see below some sugestions) and carefully interpret your correlations results.

Response 9:

We edited the discussion and added more recent sources as well as more interpretation of our finding

Comment 10:

REFERENCES

Several recent articles from IJERPH could be cited. It will you be help you to compare the results with other countries. Below I suggest some articles pre and during COVID pandemic about mental health in adolescents which may improve the introduction and discussion sections:

Response 10:

We thank the reviewer for proposing references. We have incorporated the ones below into the discussionas wellas a number of additional sources

Escobar, D.F.S.S.; Noll, P.R.S.; Jesus, T.F.; Noll, M. Assessing the Mental Health of Brazilian Students Involved in Risky Behaviors. Int. J. Environ. Res. Public Health 2020, 17, 3647.

Tahara, M.; Mashizume, Y.; Takahashi, K. Coping Mechanisms: Exploring Strategies Utilized by Japanese Healthcare Workers to Reduce Stress and Improve Mental Health during the COVID-19 Pandemic. Int. J. Environ. Res. Public Health 2021, 18, 131.

Escobar, D.F.S.S.; Jesus, T.F.; Noll, P.R.S.; Noll, M. Family and School Context: Effects on the Mental Health of Students. Int. J. Environ. Res. Public Health 2020, 17, 6042.

Reviewer 2 Report

Dear. author,

This study aims to identify the positive and negative effects of different intensities of physical activity on mental health. It is a very interesting study.

However, I would like to point out a few things that I would like to confirm.

  1. The numerical values should be unified. For example, numbers with two decimal places and numbers with three decimal places are mixed.
  2. This study has calculated the mean values of life satisfaction, WHO-5 wellbeing, MHI-5 mental health, HBSC-SLC, MVPA, and VPA by gender and age, but why is it that the study has not conducted a comparison of the mean values by gender x age?
  3. The sample size for this study is very large. Hence, if we look at the correlations, we may find an association even if the effect size is small. Due to the nature of statistics, it is necessary to mention how to look at the results when the sample size is large.
  4. table 4: **p<0.01 needs to be stated.
  5. This is the result of multiple regression analysis, but the effect size seems to be small. Please check the statistical data again.

That's all. Please consider revising and resubmit.

Best,

Author Response

Thank you for your helpful review. Below you will find our response for each of your commenys. Many thanks for taking the time to revie wthe paper. 

Comments and Suggestions for Authors

Comment 1:

This study aims to identify the positive and negative effects of different intensities of physical activity on mental health. It is a very interesting study.

Response 1:

We thank the reviewer for the kind comments  

Comment 2:

The numerical values should be unified. For example, numbers with two decimal places and numbers with three decimal places are mixed.

Response 2:

We changed the correlation coefficients presented in Table 3 and the standardized beta values in table 4 to two decimal points.

Comment 3:

This study has calculated the mean values of life satisfaction, WHO-5 wellbeing, MHI-5 mental health, HBSC-SLC, MVPA, and VPA by gender and age, but why is it that the study has not conducted a comparison of the mean values by gender x age?

Response 3:

We have now added analyses by gender and age in tables 1 and 2, as well as p values when comparing means. We have also included analyses of the interaction between age and gender. We have addressed these analyses in the results and discussion sections.

Comment 4:

The sample size for this study is very large. Hence, if we look at the correlations, we may find an association even if the effect size is small. Due to the nature of statistics, it is necessary to mention how to look at the results when the sample size is large.

Response 4:

We are using Cohen’s operational guidelines for interpreting coefficient in large sizes studies, or as he states, in a more real life context. According to Cohen (1988), in terms of the magnitude of effect size, correlation coefficients should be interpreted in such way that r = 0.10 is considered small; r=0.30 is considered medium; and r=0.50 is considered large. We have included in the discussion text to that effect, as well as used Cohen’s guidelines for interpreting our and presented guidelines for interpretation in the results

Comment 5:

table 4: **p<0.01 needs to be stated.

Response 5:

We have now added this text in table 4.  

Comment 6:

This is the result of multiple regression analysis, but the effect size seems to be small. Please check the statistical data again.

Response 6:

We re-ran the analyses to make sure that the data presented are correct. As stated in comment 4, and with the additional critical view of Cohen’s work presented by Hemphill (2003) we provided explanation on the interpretation of such effect size. While the effects seem low, within the context of this study they are very meaningful – again, as was stressed by Hemphill. We added text to that effect in the discussion.

Round 2

Reviewer 1 Report

Congrats to the authors. All my comments were addressed.

I only have one suggestion for the Tables 1 and 2. Tables should be self-explanatory. Please, include a legend with the meaning of the acronyms (MHI-5, HBSC, WHO, MVPA, VPA...) and also what test was used to give the 'p value'

Author Response

Response to review

Comment 1:

Congrats to the authors. All my comments were addressed.

Response 2:

Thank you for acknowledging our response to your previous comments.

Comment 2:

I only have one suggestion for the Tables 1 and 2. Tables should be self-explanatory. Please, include a legend with the meaning of the acronyms (MHI-5, HBSC, WHO, MVPA, VPA...) and also what test was used to give the 'p value'

Response 2:

We added legends under tables 1 and 2 with explanations of the abbreviations and with references to the statistical tests used for providing p values